# Injecting Vision Language into Autoregressive Image Generation

## Abstract

Autoregressive (AR) models have become central to modern foundation models like large language models (LLMs) and visual-language models (VLMs). Recently, AR-based approaches have extended into text-to-image generation. Although these text-to-image AR models have been trained for visual-language token interaction, they often struggle when conditioned on visual inputs. Focusing on this drawback, in this paper, we are curious about one question: *how can we inject vision information to a pre-trained AR model to ensure its output reflects visual conditions?* We answer this question with a simple yet effective solution termed InjectAR. Our key insight is that, while a pre-trained AR model cannot handle visual inputs directly, its inherent capability for visual-language interaction can indeed support visual feature extraction. Consequently, with only a few newly introduced parameters and minimal training, a pre-trained AR generation model can successfully accommodate both text and image conditions and produce visually appealing results. To manage the relationship between textual and visual inputs, we reinforce InjectAR with a hierarchical attention mechanism, which subdivides the attention scores for textual tokens into their corresponding visual components, preventing either modality from dominating the output. As the first AR model with this capability, extensive experiments show that InjectAR achieves performance on par with, or even surpasses, state-of-the-art diffusion models. Moreover, unlike diffusion models, once trained, our method has the potential for flexible control over the positions of visual objects. Our codes will be available.

## 1 Introduction

Text-to-image generation is a multimodal-involved task, which aims to generate corresponding high-quality images based on the text descriptions provided by users. Recently, large-scale models, including models based on diffusion (Ho et al., 2020; Sohl-Dickstein et al., 2015; Song et al., 2020b) and autoregression (Van Den Oord et al., 2017; Ramesh et al., 2021; Esser et al., 2021; Yu et al., 2022; Li et al., 2024b; Sun et al., 2024; Tian et al., 2024), have exhibited impressive capabilities in generating diverse and realistic images. In contrast with the continuous image representation in diffusion models (Ramesh et al., 2021; 2022; Wei et al., 2023), autoregressive (AR) image generation typically treat images as discrete tokens, mimicking the process of text modeling via a dictionary (Achiam et al., 2023; Touvron et al., 2023a; Anil et al., 2023; Touvron et al., 2023b). This kind of "next-token prediction" exhibit a promising path towards unify the representing and generation of vision and language.

Although text-to-image AR models have been extensively trained to handle interactions between visual and language tokens, they still encounter difficulties when tasked with generating outputs conditioned on visual inputs. This is particularly evident in scenarios where the alignment between the visual features and the corresponding linguistic descriptions is critical, such as in the customized generation task Ruiz et al. (2023); Gal et al. (2022). Though Li et al. (2024c) has tried to introduce pixel-level control in the "next-scale" based model, it ignores the inherent capability of unified visual-language representation as well as interaction, and cannot preserve the main concepts in conditional image. Moreover, it can not be generally applied to the vanilla AR model.

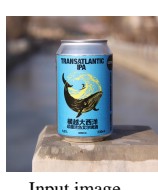
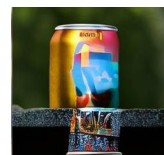

Input image · Generated based on the former gt token

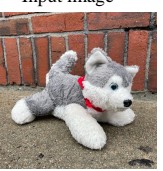
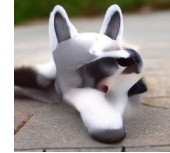

Input image · Generated based on the former gt token

Figure 1: Simply injecting the ground truth object tokens as the condition is not feasible.

In AR image generation, utilizing Vector Quantization models (Esser et al., 2021; Van Den Oord et al., 2017), images are projected into discrete tokens. These image tokens together with textual tokens then are further aligned in the autoregressive pre-training mode. This can serve as an excellent prior when dealing with vision-injecting problems. However, simply injecting the ground truth object tokens as the condition is not feasible, as shown in Figure 1. To inject this object-related vision information into the AR model, we propose a model termed InjectAR. Firstly, we utilize the BLIP to retrieve textual prompts, and the textual descriptions of the main subject and the background in the image are intentionally separated. Since the pre-trained model already learns to unify visual and textual embedding space, we straightly utilize this advantage and extract the needed feature embeddings in the image through a mask and image compacting module. Moreover, a learnable K, V mapper is attached to each attention layer, targeting at joining the visual embeddings with image generation. Furthermore, we propose a hierarchy attention component which restricts the effective region of the image condition and prevents either modality from dominating the output. Dropout and classifier-free guidance are also used in the finetuning process.

Experimental results show that our model has better prompt-fidelity as well as maintaining more object details compared to other customized models. Besides, our model has the potential for controlling the position and size of target objects. These promising results might give more inspiration and assist in building more unified autoregressive models.

Our contributions are summarized as follows:

- We are the first to introduce the discrete-type vision condition into image customized AR generation and receive superior performance.
- We design an effective vision-condition introducing framework in AR model, which is composed of text and image conditions retrieving, hierarchy attention component and the classifier-free guidance from both conditions.
- Extensive experiments are conducted, which show that our InjectAR can faithfully recover the target concept and the prompt-fidelity is fairly high. The ability of controlling object position is discussed.

## 2 RELATED WORK

In this section, we give a review on recent impressive advancements in image generation, which can be divided into two categories: diffusion models and autoregressive models. Then we summarize works related to image personalization.

### 2.1 DIFFUSION-BASED IMAGE GENERATION

Diffusion models (Ho et al., 2020; Sohl-Dickstein et al., 2015; Song et al., 2020b) regard the generation of an image as a process of gradually denoising from pure noise and are equipped with the ability by training to predict the noise applied to noisy images. Song et al. (2020a; 2023) explore

how to minimize the sampling steps. Rombach et al. (2022) models the denoising process on the latent space instead of pixel space, which compresses the image and brings strong prior. To achieve high-quality image generation and improve semantic understanding capability, large-scale diffusion models are trained with billions of image-text pairs (Saharia et al., 2022; Rombach et al., 2022; Nichol et al., 2021). Besides, there are also some other downstream models to provide additional control on the image generation (Zhang et al., 2023; Kumari et al., 2023; Gal et al., 2022; Ruiz et al., 2023; Ye et al., 2023).

## 2.2 Autoregressive image generation

The unprecedented development and incredible capability of large language models (LLMs) (Achiam et al., 2023; Touvron et al., 2023a; Anil et al., 2023; Touvron et al., 2023b; Team et al., 2023; Bai et al., 2023) exhibit a promising "next-token-prediction" path towards artificial general intelligence (AGI). To unify understanding and generation of vision and text into the same paradigm, many efforts (Van Den Oord et al., 2017; Ramesh et al., 2021; Esser et al., 2021; Yu et al., 2022; Li et al., 2024b) have been made in the field of autoregressive image generation. In contrast with the continuous image representation in diffusion models, autoregressive image generation typically treats images as discrete tokens, mimicking the process of text modeling via a dictionary.

Recently, Tian et al. (2024) treats image generation as a hierarchical multi-scale process. Li et al. (2024b) uses a masked autoregressive method and models the per-token probability distribution via a diffusion procedure. Differently, in order to exactly unify text and image modeling, LlamaGen (Sun et al., 2024) adopts the same architecture as LLM and verifies the scalability in autoregressive image generation. Our work is based on LlamaGen to verify the benefits of introducing discrete image conditions in image personalization, which eliminates the impact of different architecture design.

## 2.3 Image personalization

Aside from generating images based on textual descriptions (Ramesh et al., 2021; 2022; Saharia et al., 2022; Rombach et al., 2022; Yu et al., 2022), it is often expected to customize the appearance within the images. However, such personalization needs are typically difficult to fully describe using language. On this basis, many diffusion-based personalization approaches (Ruiz et al., 2023; Wei et al., 2023; Gal et al., 2022; Kumari et al., 2023; Ye et al., 2023) have been proposed to extract concepts from a few images and apply them to new scenarios. These methods can be divided into two parts: test-time optimization and training-based models. For test-time optimization (Ruiz et al., 2023; Gal et al., 2022; Kumari et al., 2023), the models need to be optimized each time when encountering a new concept, with the duration ranging from a few minutes to approximately one hour. For example, DreamBooth (Ruiz et al., 2023) is designed to assign the concept to a unique identifier and finetune the overall model, thus projecting image characteristics into language space. Differently, Textual Inversion (Gal et al., 2022) aims to learn a new pseudo-word (i.e., S*) bonded to the target concept. Nevertheless, the optimization procedure can be time-consuming. In contrast, some recent studies (Wei et al., 2023; Ye et al., 2023; Li et al., 2024a) are conducted to design a learnable visual encoder, which learns how to extract visual characteristics and encode these into language embedding space during training. Thus, only one forward process is needed when faced with a new image set.

However, these models are all based on diffusion models, which inherit the constraints of diffusion, i.e., enduring high inference latency and distinct paradigms with LLMs. Besides, these training-based models all require the incorporation of other pre-trained image processing model, for instance, CLIP image encoder, to map the image into continuous features and then further project into textual space. Consequently, the preservation of image characteristics are limited by the capabilities of the image encoder. Moreover, the continuous image features are inconsistent with the paradigm of language modeling, thereby impose difficulties in unifying understanding and generation between vision and language.

In contrast, our method utilize the natural discrete image tokens which unify the representing of image and text, thus being able to preserve the original image features. Furthermore, since the discrete visual and textual tokens are aligned in the autoregressive pre-training process, based on

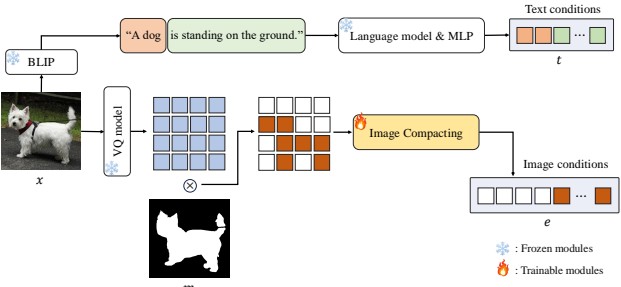

Figure 2: Retrieving text and image conditions. It is composed of two parts: designed prompts to split the main object and background & image feature extraction module.

this strong prior, we no longer need to project image features into textual space, which could reduce the computational cost and eliminate the loss of information in mapping.

## 3 METHODOLOGY

We focus on designing a customized image generation method utilizing discrete visual conditions based autoregressive models. In this section, we describe the proposed InjectAR in detail. The main architecture of the model is shown in Figure 3. The proposed InjectAR consist of three parts: retrieving text and image conditions (designed prompts to split the main object and background & image feature compacting module) (Sec. 3.1), the trainable hierarchy attention component which embed the discrete vision condition into autoregressive image generation (Sec. 3.2). Besides, random dropout and regularization loss are utilized in this finetuning process (Sec. 3.3).

### 3.1 RETRIEVING TEXT AND IMAGE CONDITIONS

Our goal is to adopt discrete visual tokens into text-to-image generation. Therefore, the first task is to retrieve textual and vision conditions. However, extracting textual descriptions for customized text-to-image generation is not trivial. To ensure the editability and the hierarchy control between text and image conditions, we intentionally separate the textual descriptions of the main subject and the background in the images, such as "A dog" and "is standing on the ground", as shown in Figure 2. This is done via BLIP (Li et al., 2022) and it helps the model distinguish the main object, especially when there exist several entities. We obtain the textual features through:

$$t = \text{MLP} \circ \text{LM}(c), \tag{1}$$

where $t \in \mathbb{R}^{L \times d}$, $L$ is the number of textual tokens, and $d$ is the dimension of textual embedding. LM is the language model used for encoding textual conditions and MLP is the projection for mapping textual conditions to visual generating space.

For the image conditions, different from previous diffusion-based customized models (Wei et al., 2023; Ye et al., 2023), we no longer rely on introducing CLIP image encoder to extract the image features. This eliminates the performance restrictions brought about by the CLIP model. In contrast, we leverage the output of the Vector-Quantized model (VQ) (Sun et al., 2024) to generate discrete visual tokens directly and use object masks to ease the impact of the redundant background. Moreover, in order to capture the most crucial information, we design an image compacting module, "IC", which consists of a 3-layer MLP to get the image conditions $e$, and we adopt the bottleneck structure for it. The inner dimension is set to 384 in training. The process of retrieving image conditions is shown as follows:

$$e = \text{IC}(m * (\text{Emb} \circ \text{VQ}(x))), \tag{2}$$

where $e \in \mathbb{R}^{N \times d}$, $N$ is the number of discrete image conditions. In training, we set it equal to the number of image tokens. And $d$ is the dimension of image embedding. $x$ and $m$ are the input image and the object mask respectively. $Emb$ is the Embedding layer of the autoregressive model.

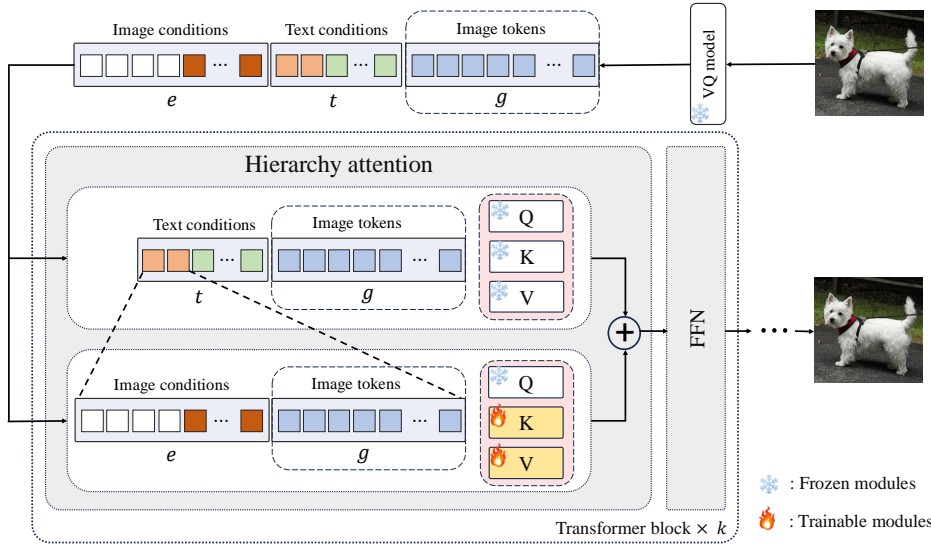

Figure 3: Hierarchy attention component. It consist of two parts: one interacts with textual conditions and the other with image conditions.

## 3.2 HIERARCHY ATTENTION COMPONENT

Usually, the visual autoregressive models utilize causal attention, which means that all the conditions as well as the tokens already generated will be attended to when generating the current token. Nevertheless, autoregressive models can become confused about how to leverage these provided image conditions effectively. Namely, during generating process, the given image conditions may exert a stronger impact in the generation of the main context. While when generating background-related tokens, the given image conditions can be distracting since these tokens to be generated are more closely tied to the textual descriptions regarding the background. In order to achieve better and effective guidance, we employ hierarchy attention component when introducing image conditions, as shown in Figure 3.

In the training process, we concat image conditions $e$, text conditions $t$ and the image tokens generated from the VQ model as the input. In the hierarchy attention component, we design to split the original causal attention module into two parts: one interacts with textual conditions and the other with image conditions.

### 3.2.1 TEXTUAL CAUSAL ATTENTION

The original causal attention is realized by a weighted sum over value features:

$$\text{Attention}(\mathbf{Q}, \mathbf{K}, \mathbf{V}, \text{Mask}) = \text{Softmax}\left(\frac{\text{Mask}(\mathbf{Q}\mathbf{K}^\top)}{\sqrt{d}}\right)\mathbf{V}, \tag{3}$$

where

$$
\begin{aligned}
\mathbf{Q} &= \mathbf{T}\mathbf{W}_q, \\
\mathbf{K} &= \mathbf{T}\mathbf{W}_k, \\
\mathbf{V} &= \mathbf{T}\mathbf{W}_v,
\end{aligned}
\tag{4}
$$

are the query, key, and values of the attention operation respectively. $\mathbf{T}$ is the concatenation of text features $t$ and image tokens $g$, i.e., $\mathbf{T} = \text{Concat}(\mathbf{t}, \mathbf{g})$ and Mask is the causal mask for "next-token prediction" task. $d$ is the dimension of image embedding. And $\mathbf{W}_q, \mathbf{W}_k, \mathbf{W}_v$ are the weights of the linear projection layers. We adopt this for interacting with text conditions.

### 3.2.2 HIERARCHY CAUSAL ATTENTION

With regard to the interaction with image conditions, we decouple this into another causal-attention module. Moreover, in order to encourage the model to selectively prioritize image conditions when

generating object-specific tokens, we introduce a hierarchical attention adjustment mechanism, utilizing the attention score $\text{Hier} \in \mathbb{R}^{N \times 1}$ derived from label-specific descriptions, where $N$ is the number of image tokens. This hierarchy causal attention module can be expressed as:

$$\text{Attention}(\mathbf{Q}', \mathbf{K}', \mathbf{V}', \text{Mask}', \text{Hier}) = \text{Softmax}\left(\frac{\text{Hier} \circ \text{Mask}'(\mathbf{Q}'(\mathbf{K}')^{\top})}{\sqrt{d}}\right)\mathbf{V}', \qquad (5)$$

where

$$\begin{aligned}
\mathbf{Q}' &= \text{Concat}(\mathbf{e}, \mathbf{g})\mathbf{W}_q, \\
\mathbf{K}' &= \text{Concat}(\mathbf{e}\mathbf{W}'_k, \mathbf{g}\mathbf{W}_k), \\
\mathbf{V}' &= \text{Concat}(\mathbf{e}\mathbf{W}'_v, \mathbf{g}\mathbf{W}_v),
\end{aligned} \qquad (6)$$

are the query, key, and values of the attention operation respectively. $\text{Mask}'$ is the causal mask for image conditions and visual tokens, and $\mathbf{W}'_k$, $\mathbf{W}'_v$ are the weights of the trainable linear projection layers. In order to speed up convergence, we initialize $\mathbf{W}'_k$ as well as $\mathbf{W}'_v$ from $\mathbf{W}_k$ and $\mathbf{W}_v$. Then the two causal attention module are fused:

$$\text{ATTN} = \text{Attention}(\mathbf{Q}, \mathbf{K}, \mathbf{V}, \text{Mask}) + \lambda \cdot \text{Attention}(\mathbf{Q}', \mathbf{K}', \mathbf{V}', \text{Mask}', \text{Hier}), \qquad (7)$$

where $\lambda$ is configured as a hyperparameter and is set to 1 during training (The absent portions are filled with zeros). Thus we could control the injection of precise image conditions into the image generating process based on the attention area of label-specific descriptions. Besides, it also restricts the effective region of the image condition and reduces the chances of positional confusion between objects generated from the image condition and locations guided by the language.

## 3.3 GUIDANCE FROM BOTH CONDITIONS

During training, only the weights of image compacting module (IC) and the trainable linear projection of image conditions $\mathbf{W}'_k$, $\mathbf{W}'_v$ are optimized. The adopted training objective is:

$$\mathcal{L}_{overall} = \mathcal{L}_{ce} + \lambda_{reg} \cdot \mathcal{L}_{reg}, \qquad (8)$$

where $\mathcal{L}_{ce}$ is the Cross-Entropy loss for the original training and $\mathcal{L}_{reg}$ is the regularization loss on the image values $\mathbf{e}\mathbf{W}'_v$:

$$\mathcal{L}_{reg} = ||\mathbf{e}\mathbf{W}'_v||_1. \qquad (9)$$

We also random drop 10% image conditions in training so that we could enable classifier-free guidance for both conditions. In inference, the logits $\tilde{l}(\mathbf{t}, \mathbf{e})$ is formulated by:

$$\begin{aligned}
\tilde{l}(\mathbf{t}, \mathbf{e}) = \ &l(\varnothing, \varnothing) \\
&+ s_t \cdot (l(\mathbf{t}, \varnothing) - l(\varnothing, \varnothing)) \\
&+ s_e \cdot (l(\mathbf{t}, \mathbf{e}) - l(\mathbf{t}, \varnothing)),
\end{aligned} \qquad (10)$$

where $s_t$ is the textual scale of classifier-free guidance and $s_e$ is the visual scale. We could adjust the scales in the inference stage.

## 4 EXPERIMENTS

### 4.1 EXPERIMENTAL SETTINGS

#### 4.1.1 DATASETS

To train the proposed InjectAR, we utilize the testset of OpenImages (Kuznetsova et al., 2020) as our training dataset. It contains 125k images with 600 object classes, associated with bounding box annotations, object masks and corresponding labels. Following Wei et al. (2023), we filtered the data by region size and aspect ratio, selecting about 47k images for training. Textual descriptions are generated using BLIP, with the prompt of corresponding labels. During training, the image is cropped according to the bounding box annotations and resized to $256 \times 256$. Object masks are also used to extract foreground image features.

For inference, we simply adopt the concept images and subject masks from Wei et al. (2023), which contain 20 objects.

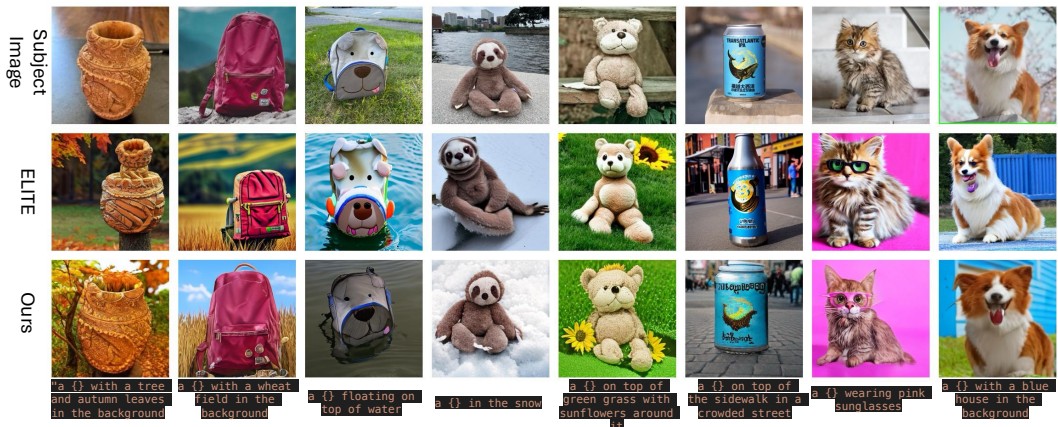

Figure 4: Comparison with existing methods. The rows are original, ELITE and ours respectively.

### 4.1.2 IMPLEMENTATION DETAILS

Following Sun et al. (2024), we utilize pre-trained FLAN-T5 XL (Chung et al., 2024) as the text encoder, and precompute text embeddings of text descriptions generated by BLIP. During training, text embeddings are left-padded and the maximum length is 120. We utilize the pre-trained Vector-Quantized model (VQ) from Sun et al. (2024) with a downsampling rate of 16. Our autoregressive generation model is based on the pre-trained text-to-image stage-I model from Sun et al. (2024) During training, the learning rate is set to $5e - 5$. We adopt AdamW optimizer with $\beta_1 = 0.9$, $\beta_2 = 0.999$, and weight decay is set to 0.01. Total batch size is 16, and $\mathcal{L}_{reg}$ is set to 0.01. Random resize, crop and rotation is employed in the image conditions.

### 4.1.3 EVALUATION METRICS

For quantitative evaluation, we adopt three metrics: CLIP-T, CLIP-I, and DINO-I as in Ruiz et al. (2023). The editing prompts set from Ruiz et al. (2023) are adopted, which contains 25 editing prompts for non-live objects and live objects separately. We randomly generated 5 images for each object-prompt pair, ultimately producing 2500 images in total. CLIP-T is defined as the cosine similarity of CLIP embeddings between text prompts and the generated images, which conveys prompt fidelity. While for CLIP-I, we calculate the average cosine similarity of CLIP visual embeddings between the generated and concept images, which indicates the subject fidelity. DINO-I is the average cosine similarity between the ViT-S/16 DINO (Caron et al., 2021) embeddings of generated and real images, and it concentrates more on structural details. More details about the inference set and editing prompts can be found in the *Suppl*.

### 4.2 COMPARISON WITH EXISTING METHODS

We compare our results with other existing methods in Figure 4. Note that other models are all based on diffusion. Our model achieves excellent object-fidelity and edibility compared with others.

### 4.3 QUANTITATIVE RESULTS

Moreover, we conducted quantitative evaluations to validate the performance of our proposed InjectAR compared with other diffusion-based methods (Note that these results for test-time optimization models are all based on finetuning on a single image for fair comparison). As shown in Table 1, our model achieves the best prompt-alignment score, which demonstrates its remarkable editability. And on metric DINO-I which concentrates on details consistency, we also obtain superior performance, showing that our model is capable of preserving more detailed information compared to other models. Besides, our model achieves comparable performance on the CLIP-I metric, which exhibits excellent ability to generate high-quality images.

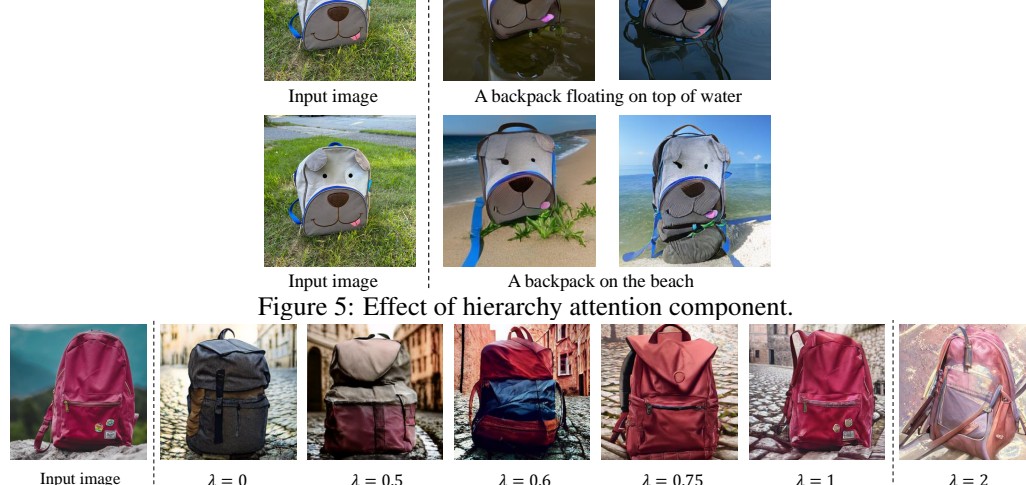

Figure 5: Effect of hierarchy attention component.

Figure 6: Impact of hierarchy attention scalar $\lambda$ in inference. The editing prompt is "A backpack on a cobblestone street". When increasing $\lambda$, the main content is steering towards the given condition.

### 4.4 EMPIRICAL STUDY

#### 4.4.1 EFFECT OF HIERARCHY ATTENTION COMPONENT

We conducted an ablation study on the effect of the hierarchy attention component. As shown in Figure 5, without this hierarchy attention component, the generated images are more prone to exhibit overlapping objects, which results from the mutual interference between the generation processes guided by visual and textual inputs. Our design successfully alleviates the incidence of this issue.

#### 4.4.2 IMPACT OF HIERARCHY ATTENTION SCALAR

As shown in Figure 6, when hierarchy attention scalar $\lambda = 0$, the generated image is barely influenced by the image condition in inference. With the increase of $\lambda$, the impact from the image condition increases. When utilizing $\lambda = 1$, we arrive at a fairly satisfactory result. However, a much larger $\lambda$ can introduce unreasonable image information, which may destroy the whole image.

#### 4.4.3 IMPACT OF VISUAL SCALE

Visual scale $s_e$ can be adjusted in the inference stage, which controls the image generation direction between textual and visual conditions. We set $s_t = 7.5$ by default. From Table 2, we can see that with the increase of $s_e$, the text-fidelity decreases, while the image-fidelity metrics CLIP-I and DINO-I increase at first. When the visual scale $s_e$ is too large, this damages the meaning and quality of the whole image, thus the image-fidelity metrics decrease.

## 5 DISCUSSIONS

In our experiments, we observed highly promising results that may serve as a potential direction for future research. With adjusted augmentation, the model demonstrates the ability to specify object locations. As shown in Figure 7a, we could potentially control the position of the object by modifying the input image conditions, which is not feasible for diffusion-based models (Wei et al., 2023; Kumari et al., 2023; Ruiz et al., 2023). Moreover, the generated objects are also well-integrated with the surrounding environment.

Even though our InjectAR achieves excellent performance in generating high-quality and fine-grained images, it still inherits several limitations from the generation model. In some situations, the language descriptions were not generated effectively or disrupted by the image conditions. For

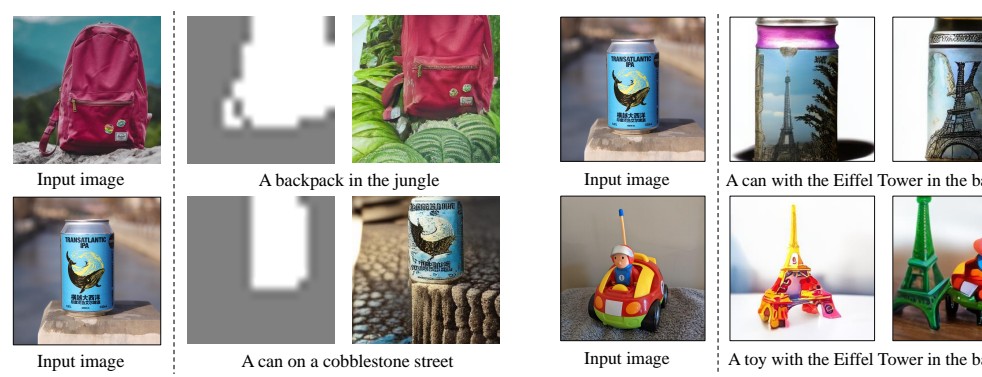

(a) By modifying the image conditions, we could potentially control the position of the object based on the AR image generation model, which is barely feasible in diffusion.

(b) Our model still inherits several limitations from the generation model. In some situations, the language descriptions were not generated effectively or disrupted by the image conditions.

Figure 7: More discussion results.

Table 1: Quantitative comparisons with existing methods. The best results are in **bold**.

| Method | CLIP-T ($\uparrow$) | CLIP-I ($\uparrow$) | DINO-I ($\uparrow$) |
|---|---|---|---|
| Textual Inversion (Gal et al., 2022) | 0.183 | 0.663 | 0.462 |
| DreamBooth (Ruiz et al., 2023) | 0.251 | 0.785 | 0.674 |
| Custom Diffusion (Kumari et al., 2023) | 0.245 | **0.801** | 0.695 |
| ELITE (Wei et al., 2023) | 0.255 | 0.762 | 0.652 |
| Ours | **0.290** | 0.769 | **0.722** |

Table 2: Impact of visual scale $s_e$. All these metrics are calculated under $\lambda = 1$.

| Method (visual scale) | CLIP-T ($\uparrow$) | CLIP-I ($\uparrow$) | DINO-I ($\uparrow$) |
|---|---|---|---|
| $s_e = 1.5$ | **0.314** | 0.731 | 0.635 |
| $s_e = 5$ | 0.290 | 0.769 | 0.722 |
| $s_e = 7$ | 0.283 | **0.770** | **0.725** |
| $s_e = 10$ | 0.275 | 0.764 | 0.721 |
| $s_e = 13.5$ | 0.270 | 0.760 | 0.720 |

instance, in Figure 7b, the Eiffel Tower which should have appeared in the background, was mistakenly generated within the can. In the bottom row, The tower's color and material properties were altered by the image conditions, resulting in a green plastic appearance, and in some instances, the features completely fused. We consider this as a valuable problem to be addressed in future studies.

## 6 CONCLUSION

In this paper, a new and effective vision-condition introducing framework in AR model is proposed. In contrast with the continuous image features, our method utilizes the natural discrete image tokens which unify the representing of image and text, thus being able to preserve the original image features. Our model consists of text and image conditions retrieving, hierarchy attention component and the design of classifier-free guidance from both conditions. During training, only the weights of the image compacting module and the trainable linear projection of image conditions are optimized. Experiments show that our model achieves the best prompt-alignment performance, which demonstrates its remarkable editability. The qualitative and quantitative comparisons with other models show its superior capability to retain the details and generate high-quality images.

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

# A APPENDIX

## A.1 EDITING PROMPTS

Following Wei et al. (2023); Kumari et al. (2023); Gal et al. (2022), we utilize the editing prompts set from Ruiz et al. (2023), which contains 25 editing prompts for non-live objects and live objects separately, as shown below (Note that the $S_*$ is substituted for the corresponding labels in inference):

For non-live objects:

- "a $S_*$ in the jungle",
- "a $S_*$ in the snow",
- "a $S_*$ on the beach",
- "a $S_*$ on a cobblestone street",
- "a $S_*$ on top of pink fabric",
- "a $S_*$ on top of a wooden floor",
- "a $S_*$ with a city in the background",
- "a $S_*$ with a mountain in the background",
- "a $S_*$ with a blue house in the background",
- "a $S_*$ on top of a purple rug in a forest",
- "a $S_*$ with a wheat field in the background",
- "a $S_*$ with a tree and autumn leaves in the background",
- "a $S_*$ with the Eiffel Tower in the background",
- "a $S_*$ floating on top of water",
- "a $S_*$ floating in an ocean of milk",
- "a $S_*$ on top of green grass with sunflowers around it",
- "a $S_*$ on top of a mirror",
- "a $S_*$ on top of the sidewalk in a crowded street",
- "a $S_*$ on top of a dirt road",
- "a $S_*$ on top of a white rug",
- "a red $S_*$",
- "a purple $S_*$",
- "a shiny $S_*$",
- "a wet $S_*$",
- "a cube shaped $S_*$".

For live objects:

- "a $S_*$ in the jungle",
- "a $S_*$ in the snow",
- "a $S_*$ on the beach",
- "a $S_*$ on a cobblestone street",
- "a $S_*$ on top of pink fabric",
- "a $S_*$ on top of a wooden floor",

- "a $S_*$ with a city in the background",
- "a $S_*$ with a mountain in the background",
- "a $S_*$ with a blue house in the background",
- "a $S_*$ on top of a purple rug in a forest",
- "a $S_*$ wearing a red hat",
- "a $S_*$ wearing a santa hat",
- "a $S_*$ wearing a rainbow scarf",
- "a $S_*$ wearing a black top hat and a monocle",
- "a $S_*$ in a chef outfit",
- "a $S_*$ in a firefighter outfit",
- "a $S_*$ in a police outfit",
- "a $S_*$ wearing pink glasses",
- "a $S_*$ wearing a yellow shirt",
- "a $S_*$ in a purple wizard outfit",
- "a red $S_*$",
- "a purple $S_*$",
- "a shiny $S_*$",
- "a wet $S_*$",
- "a cube shaped $S_*$".

