# OpenReview forum: "Injecting Vision Language into Autoregressive Image Generation"
_ICLR.cc/2025/Conference — ICLR 2025 Conference Withdrawn Submission_

### Official Review · Reviewer_e983 · 2024-10-30

**Soundness:** 2
**Presentation:** 3
**Contribution:** 2
**Rating:** 5
**Confidence:** 4

**Summary:**

The proposed InjectAR, injects visual information into a pre-trained autoregressive model. To make sure the model's output reflects visual conditions, InjectAR designs a hierarchical attention mechanism, which maps attention scores from text tokens to their corresponding visual components. The method introduces only a few new parameters, requiring minimal additional training, and enables AR models to work effectively with both text and visual inputs.

**Strengths:**

InjectAR introduces discrete vision conditions into pre-trained AR models, achieving superior performance compared to traditional methods like diffusion models.It employs a hierarchical attention mechanism and classifier-free guidance to effectively combine text and visual inputs, ensuring both modalities contribute without dominance.InjectAR requires minimal additional training while delivering high prompt fidelity and flexible control over object placement, matching or surpassing state-of-the-art models.

**Weaknesses:**

1. **Image Quality and Subject Consistency**: As shown in Figure 4, the generated images are not visually appealing, and maintaining consistency between the generated objects and those in the subject images remains challenging.
2. **Reliance on BLIP for Text Encoding**: In the implementation details and Figure 2, the authors mention that they “utilize the pre-trained FLAN-T5 XL as the text encoder and precompute text embeddings from descriptions generated by BLIP.” However, since BLIP is an older image captioning model, its reliability may be insufficient, potentially limiting the quality of the text-conditioned image generation.
3. **Limited Capability in Handling Multiple Objects**: The input images used in the experiments contain only a single object, making it unclear how well the model would perform when processing images with multiple objects.

**Questions:**

1. Why is it difficult to maintain consistency between the generated objects and those in the subject images?
2. Given that more advanced image captioning models are available, why did you choose BLIP as the image captioner?
3. Can you provide examples or cases that demonstrate the model’s performance when handling images containing multiple objects?

---

### Official Review · Reviewer_PsJT · 2024-11-03

**Soundness:** 3
**Presentation:** 1
**Contribution:** 1
**Rating:** 3
**Confidence:** 4

**Summary:**

This paper aims to introduce a discrete-type vision condition into AR model for image generation. A text-image condition retrieving and a hierarchy attention component are leveraged. Some experiments are conducted to demonstrate the effectiveness.

**Strengths:**

- The paper contains some experiments.
- The paper provides some implementation details.

**Weaknesses:**

- The proposed method lacks novelty. Using BLIP to generate prompts is common.
- The proposed attention mechanism and classifier-free guidance are commonly used.
- The paper is not well-presented such as an empty line in Line 211, section 3.2 is very confusing, and "Suppl" in Line 361.
- More up-to-date methods should be compared in Table 1.
- Experimental results are very limited.
- The layout of Fig 5 and Fig 6.

**Questions:**

NA

---

### Official Review · Reviewer_TbD6 · 2024-11-03

**Soundness:** 2
**Presentation:** 3
**Contribution:** 2
**Rating:** 3
**Confidence:** 3

**Summary:**

This paper presents a method for fine-tuning an auto-regressive (AR) image generation model to allow customization based on single-object images, using object masks as ground truth labels. First, text and image features are extracted from the input image using primarily frozen models. For text conditioning, BLIP is employed with an inductive bias to separate object and background information; for visual conditioning, pre-trained VQ models generate patch-wise discrete tokens, which are processed through the object mask and then pass through a 3-layer trainable MLP (referred to as the “Image Compacting module”). Second, to enable the model to effectively leverage these conditions, a hierarchical attention mechanism is introduced: an additional token-wise weight, $Hier$, is integrated into the attention blocks for visual conditioning. Experimental results indicate that the proposed model achieves state-of-the-art performance on the CLIP-T and DINO-I metrics.

While this paper introduces a novel approach for training AR models to incorporate image conditions, I find that many components of its model design lack sufficient justification, both in terms of motivation and experimental validation. Please read the weakness section for details. In its current form, I do not believe it is ready for publication at this conference.

**Strengths:**

1. The paper is generally clear and well-structured. The methodologies easy to follow; the experiments presented with clarity.

2. In both generating textual conditions and visual conditions (the latter using object masks), the heuristic of “separating objects from the background” is applied. This heuristic is intuitively useful in tasks requiring *object* customization, as it allows for more targeted conditioning on specific elements within an image.

**Weaknesses:**

I am not entirely persuaded by some components of the model design.
1. Given the robust visual condition extraction pipeline, I question the necessity of extracting textual conditions for this "image in, image out" task. Specifically, in the visual condition pipeline, there is already a pre-trained VQ model, an object mask, and a trainable MLP, which together provide a strong framework for extracting visual representations. At the very least, I would expect the authors to clarify their motivation for including textual conditions and to present experimental results demonstrating how these conditions contribute.

2. The hierarchical attention mechanism is not explained in sufficient detail. Specifically, it is unclear how $Hier$ is derived, which seems essential to the new hierarchical mechanism. Aside from a brief mention of it being “derived from label-specific descriptions,” there is little explanation. Additionally, $Hier$ is not explicitly shown in Figure 3, which depicts the model pipeline.

3. I think the “hierarchical attention mechanism” is two standard attention mechanisms, with the visual attention values weighted according to their relevance to the textual features of objects. Although the mechanism itself is straightforward, this specific design choice lacks clear motivation. Why not apply the same weighting mechanism to the attention of textual conditions? And why are “the textual features of objects” used as a target to guide $Hier$ (based on my understanding from the dotted line in Figure 3) rather than, for example, “the visual patches after the object masks”?

The experimental results are insufficient to justify the model design, and as a result, many components seem arbitrary.

4. If the authors claim novelty in their textual & visual condition extraction components, I would expect more ablation studies on these. For instance, what if only the visual pipeline is used? What if a pre-trained VAE is used instead of vector quantization?

5. If the authors claim novelty in their hierarchical attention component, I would like to see more ablation studies on that as well. Some examples are mentioned in point 3.

**Questions:**

I listed all my questions with the weaknesses.

---

### Official Review · Reviewer_jQMa · 2024-11-04

**Soundness:** 3
**Presentation:** 4
**Contribution:** 3
**Rating:** 3
**Confidence:** 4

**Summary:**

This paper introduces InjectAR, a new method for injecting discrete visual tokens into autoregressive (AR) image generation models, enabling effective integration of multimodal conditions. InjectAR addresses the challenge of incorporating visual information as conditioning in traditional AR text-to-image generation, by leveraging a hierarchical attention mechanism.

**Strengths:**

- Introduces a new method to integrate discrete vision tokens in autoregressive image generation models.
- Propose a new hierarchical attention mechanism for balanced multimodal customized control with low-cost fine-tuning.
- Well-organized story and easy to follow.

**Weaknesses:**

- Limited motivation, many current image-generation models, such as ControlNet, already support both image and text as condition.
- Insufficient experiments and analysis. The generation results lack strong metrics and thorough analysis to convincingly support the proposed approach.
- Incomplete comparison with existing methods, please conduct a more comprehensive literature review.

**Questions:**

- Please include additional evaluation metrics, such as the MSCOCO FID, to assess image quality. Visually, it appears that visual quality is compromised for the added control signals in Figure 4; showing images generated by the base model would also be helpful.
- Could you clarify why the CLIP-I similarity is lower than that of other methods?
- Techniques like ControlNet can achieve similar results using a mask as an additional input at a lower training cost and minimal architecture modification. Please provide a comparison between your method and ControlNet.

---

### Note · Authors · 2024-11-19

I have read and agree with the venue's withdrawal policy on behalf of myself and my co-authors.